# Task adaptation by biologically inspired stochastic comodulation

## Abstract

Brain representations must strike a balance between generalizability and adaptability. Neural codes capture general statistical regularities in the world, while dynamically adjusting to reflect current goals. One aspect of this adaptation is stochastically co-modulating neurons' gains based on their task relevance. These fluctuations then propagate downstream to guide decision making. Here, we test the computational viability of such a scheme in the context of multi-task learning. We show that fine-tuning convolutional networks by stochastic gain modulation improves on deterministic gain modulation, achieving state-of-the-art results on the CelebA dataset. To better understand the mechanisms supporting this improvement, we explore how fine-tuning performance is affected by architecture using Cifar-100. Overall, our results suggest that stochastic comodulation can enhance learning efficiency and performance in multi-task learning, without additional learnable parameters. This offers a promising new direction for developing more flexible and robust intelligent systems.

## 1 Introduction

The perception of the same sensory stimulus changes based on context. This perceptual adjustment arises as a natural trade-off between constructing reusable representations that capture core statistical regularities of inputs, and fine-tuning representations for mastery in a specific task. Gain modulation of neuronal tuning by attention can boost task-informative sensory information for downstream processing (Maunsell, 2015). It has also served as biological inspiration for some of the most successful machine learning models today (Vaswani et al., 2017; Brown et al., 2020; Touvron et al., 2023). However, modeling results suggest that deterministic gain modulation loses some of its ability to highlight task-relevant information as it propagates across processing layers and consequently has limited effects on decisions made downstream (Lindsay & Miller, 2018). Additionally, some have argued based on experimental data in humans that the behavioral benefits of attention may largely come from effective contextual readouts rather than encoding effects (Pestilli et al., 2011). Thus, neural mechanisms underlying context dependent sensory processing remain a topic of intense investigation (Ferguson & Cardin, 2020; Shine et al., 2021; Naumann et al., 2022; Rust & Cohen, 2022; Zeng et al., 2019).

A less well known mechanism for task adaptation of neural responses has received recent experimental support. It involves modulating the variability of neural responses in a task-dependent manner. Experimentally, it has been observed that responses of neurons in visual areas of monkeys exhibit low-dimensional comodulation (Goris et al., 2014; Rabinowitz et al., 2015; Bondy et al., 2018; Huang et al., 2019; Haimerl et al., 2021). This modulation occurs at a fast time scale, affects preferentially neurons that carry task-relevant information, and propagates in sync with the sensory information to subsequent visual areas (Haimerl et al., 2021). Related theoretical work has proposed that such task-dependent comodulation can serve as a label to facilitate downstream readout (Haimerl et al., 2019), something which was later confirmed in V1 data (Haimerl et al., 2021). Finally, incorporating targeted comodulation into a multilayer neural network enables data efficient fine-tuning in single tasks. Such fine-tuning allows the network to instantly revert back to the initial operating regime once task demands change, eliminating any possibility for across-task interference, and can outperform traditional attentional neural mechanisms based on gain increases (Haimerl et al., 2022).

Although stochastic comodulation has shown potential for effective task fine-tuning, it has only been tested using simple networks and toy visual tasks. Furthermore, the neural mechanisms that determine the appropriate pattern of comodulation for any given task remains unclear. Here we ask "What kind of computations can stochastic gain modulation support?" and "What kind of architecture is required for context dependence across tasks?" Specifically, we incorporate stochastic modulation into large image models and optimize a controller subcircuit to determine the comodulation pattern as a function of the task. Our solution improves on deterministic gain modulation and surpasses state-of-the-art task fine-tuning on the CelebA dataset. We use the simpler Cifar100 and vary network architecture to explore different scenarios in which comodulation is more or less beneficial. We characterize features of the resulting network embeddings to better understand how and when comodulation is beneficial for task adaptation. We find that although comodulation does not always improve on deterministic attention, it is at least comparable with it. Moreover, the comodulation-based solution always provides better more accurate measures of output confidence, even in scenarios without a clear performance improvement. Overall, our results argue for comodulation as a computationally inexpensive way of improving task fine-tuning, which makes it a good candidate for the context dependent processing of sensory information in the brain.

## 2 RELATED WORK

Multi-task learning can take many forms. For instance, meta-learning approaches such as MAML aim to create models that are easy to fine tune, so that a new task can be learned with few training samples (Finn et al., 2017). Here, we follow the formulation of Caruana (1997). Given an input data distribution, $p(\mathbf{x})$, a task corresponds to a rule for mapping inputs $\mathbf{x}$ into outputs $\mathbf{y}$, as specified by a loss function $\mathcal{L}$. The goal of multi-task learning is to harness interdependencies between tasks to build good representations of the inputs; this leads to performance improvements when learning them simultaneously as opposed to treating them in isolation. This can be achieved through architecture design, task relationship learning, or optimization strategies (Crawshaw, 2020).

Architecture design involves building upon a common baseline across tasks by utilizing a shared backbone. This backbone generates feature representations rich enough so that task adaptation only needs to adjust a readout layer, sometimes referred to as the output head (Zhang et al., 2014). During training, simple adjustments to the shared network, such as layer modulation (Zhao et al., 2018) or feature transformations, can additionally be applied (Strezoski et al., 2019; Sun et al., 2021) to encourage reusable representations. More involved architectures are also possible, for instance, tasks can have additional separate attention modules, comprised of convolutional layers, batch norm (Ioffe & Szegedy, 2015) and a ReLU non-linearities (Liu et al., 2019). These additional components have separate parameters, incurring costs that grow with the number of tasks. In contrast, our controler provides a compact constant size parametrization that is shared across task.

Our solution falls into the broad category of network adaption approaches (Mallya & Lazebnik, 2018; Mallya et al., 2018; Zhang et al., 2020), where the starting point is a base network pretrained extensively on one large task. The parameters of the resulting network are then frozen and a separate (smaller) set of parameters that manipulate the network's features, in our case stochastic gain parameters, are optimized to improve performance on one different task. In previous work this process has the goal of fine-tuning to individual new tasks, whereas our approach extends this process for an entire task family. In all cases, fine-tuning a backbone comes at a low computational cost, as changes needed for the new task are local and specified by few tunable parameters. The data distribution in the second task can differ either just in the output labels, or in both the input distribution and its map into outputs.

More generally, gating mechanisms have been a staple in machine learning for a long time, in LSTMs(Hochreiter & Schmidhuber, 1997), which use of inputs and forget gates, but also in CNNs (Dauphin et al., 2017; Van den Oord et al., 2016). Furthermore, multiplicative interactions (Jayakumar et al., 2020) such as our gain modulation are at the core of many of today's succesful neural network architectures, e.g. Transformers (Vaswani et al., 2017). Closer to our work, are feature transformation architectures such as the conditioning layer (Perez et al., 2018) which contextualizes the features of convolutional layers by doing channel-wise scaling and additions, using context weights generated by a natural language processing method. What is unique to our approach is the stochastic nature of the gains and how the variability is used to affect downstream readouts.

## 3 METHODS

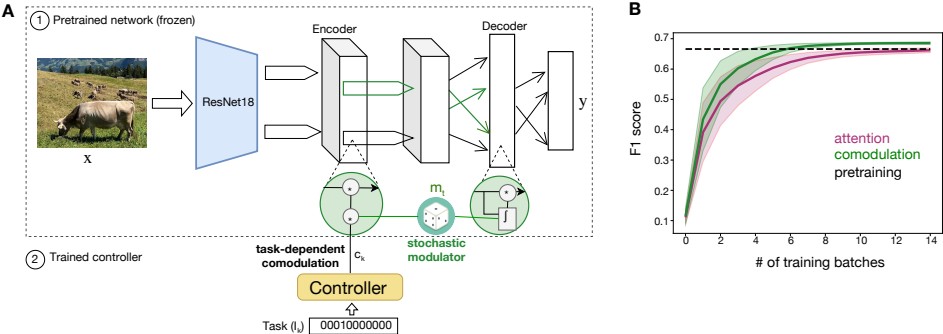

Figure 1: Task fine-tuning by stochastic comodulation. **A)** Schematic of the architecture: ResNet18 backbone with 2 additional convolutional layers one MLP decoding layer. Encoding layer gains are stochastically modulated by $m_t$, sampled from a normal distribution, with a task-dependent pattern of covariability determined by the Controller. The decoding layer gains adjust as a function of the correlations between individual neuron activities and the same modulator. **B)** Evolution of fine-tuning classification performance over learning for stochastic gain comodulation and deterministic gain modulation (attention) for CelebA multi-task classification.

**Fine-tuning by comodulation.** We incorporate the idea of a stochastic comodulation-based readouts from Haimerl et al. (2019; 2022) in a large image classification architecture to perform conditional network adaptation (Fig. 1A). The model involves a strong feature extractor, here ResNet18 (He et al., 2016) pretrained on ImageNet (Deng et al., 2009), followed by 2 convolutional layers, the first of which is the encoder, followed by a MLP layer serving as the decoder, and finally the decision layer. The general idea is that a one-dimensional i.i.d. gaussian noise source $m_t$, which we will refer to as the modulator, is projected into the encoder layer via a task-specific map provided by the controller, and multiplicatively changes the activity of neurons in that layer. These fluctuations in gain propagate through the subsequent layers of the network to the decoder. This converts the question of which neurons within the decoding layer carry the most task relevant information into the simpler question of which neurons of the encoding layer co-fluctuate the most with the modulator $m_t$. In the original version of this idea the correct pattern of co-modulation, targeted towards task relevant neurons in the encoder was either set by hand, or learned independently for each new task. Here the controller module aims to learn a parametric solution for an entire family of tasks.

More concretely, the encoder layer activity transforms the outputs of the feature extractor as:

$$\boldsymbol{h}_{kt}^{l} = \text{rectifier}(\boldsymbol{W} \circledast \boldsymbol{h}^{l-1} + \boldsymbol{b}) \odot (m_t \boldsymbol{c}_k), \tag{1}$$

where $\circledast$ denotes the convolution operator, $\boldsymbol{W}$ and $\boldsymbol{b}$ are the layer's weights and biases, and $\boldsymbol{c}_k$ are the context weights for task $k$. Given the original theory, these are expected to be large for task-informative neurons and close to zero for uninformative ones. The context weights are generated by a small controller sub-network, which maps individual tasks into context weights. The stochastic signal $m_t$ is sampled i.i.d. from $|\mathcal{N}(0, 0.4)|$; this variance is not so large that it drowns the visual signal, but large enough to produce interesting fluctuations in the neural responses. Each input is propagated from the encoder to the decoder a total of $T$ times, each time with different random draw of modulator $m_t$.

The decoder is the last layer before the decision, i.e. its activities are linearly read out to produce the network output. Task relevance is determined based on the degree of correlation between that individual activations and the modulator; this determines a set of gains, which multiplicatively modulate the layer's outputs, thus affecting the readout. The decoder gains follow the general formulation of stochastic comodulation from Haimerl et al. (2022), with a few changes. Given decoder layer activities $\boldsymbol{h}_{kt}^{J}$, the gain is computed as

$$\boldsymbol{g}_k = \sum_t^T \bar{m}_t \bar{h}_{kt}^{(J)} \tag{2}$$

where $\bar{m}_k$ is the mean-substracted modulator, and $\bar{h}_t^{(J)}$ the corresponding mean-subtracted decoder layer activity; the result is then normalized to be in the range [0,1], using min-max normalization, and used as:

$$h_k^J = g_k \text{rectifier}(W_J h_k^{J-1} + b) \tag{3}$$

where $h_k^{J-1}$ denotes the context-dependent input to the decoder. The presence of gains $g_{nt}$ shifts the embedding of inputs to adaptively change the network outputs, despite using the same readout weights. When assessing test performance, we use unit gain in the encoder (no encoding noise) paired with the estimated output gains in the decoder layer.

**Controller.** To encode the tasks and produce task-dependent context weights, we use a two-layer MLP neural network as the controller. This receives as input a one-hot encoding of the current task, and outputs a vector $c_k$ of the same dimensionality as the numbers of channels in the convolutional layer (i.e. $c_k$ entries are identical for all neurons within a channel). This can be expressed as:

$$c_k = \text{Controller}(I_k), c_k \in \mathbb{R}^{C \times 1 \times 1} \tag{4}$$

where C is the number of channels in the encoder layer.

**Training procedure.** We first pretrain the base network on a primary task, then fine-tune it on a family of tasks related to the first one. During fine-tuning, the weights of the network are frozen and the parameters of the controller, which determine the coupling weights, are optimized using Adam (Kingma & Ba, 2014). When fine-tuning, we have the option to include or exclude the gain variability during the training process. Training without gain fluctuations is computationally convenient as the same controller can be used for both deterministic and stochastic modulation, but including it can make fine-tuning more data-efficient in some setups. More detailed considerations with regards to each experiment follow below.

## 4 EXPERIMENTS

We demonstrate the effects of using comodulation in a series of numerical experiments in two datasets 1) a multi-task learning setup with the CelebA dataset (Liu et al., 2015) and 2) the CIFAR-100 dataset (Krizhevsky et al., 2009), where we use the superclasses as a task indicator. The "attention" baseline uses deterministic gain modulation defined by the controlled for the encoding layer, but with no additional effects on the decoder (i.e. decoder gains 1).

### 4.1 ATTRIBUTE CLASSIFICATION

**Setup.** CelebA (Liu et al., 2015) is a common large-scale multi-task image classification database containing images of faces and labels for 40 attributes for each image. Each task involves a binary classification of an attribute, for example classifying whether the person wears glasses or if they are smiling. In this experiment, we pretrain and fine-tune the network on the same tasks, i.e. classifying the 40 attributes. When fine-tuning, we only optimize the controller's parameters and keep every other parameter in the network fixed, including the weights of the decision layer. For pretraining, we use a batch size of 256, with a learning rate of 0.0002. We then fine-tune with a batch size of 64 and a learning rate of 0.02. These learning rates were found by a grid search hyperparameter tuning. For every experiment, we report averaged results over five seeds.

**Results.** In Table 1, we compare the comodulation to the previous state-of-the-art methods, measured as the average relative improvement over a common baseline, $\Delta_p$. In line with previous literature, this baseline is provided by the hard sharing method, which entails jointly learning every task with vanilla optimizers and sharing all parameters until the decision layer. This metric is commonly used in multi-task learning (Ding et al., 2023; Vandenhende et al., 2021), in the case of CelebA it is computed as:

$$\Delta_p = 100\% \times \frac{1}{N} \sum_{n=1}^{N} \frac{(-1)^{p_n}(M_n - M_n^{\text{baseline}})}{M_n^{\text{baseline}}},$$

where $N$ is the number of metrics used for the comparison, here $N = 3$: recall, precision and F1-score; $M_n$ is the value of the $n - th$ metric, while $M_n^{\text{baseline}}$ is the corresponding vale for the

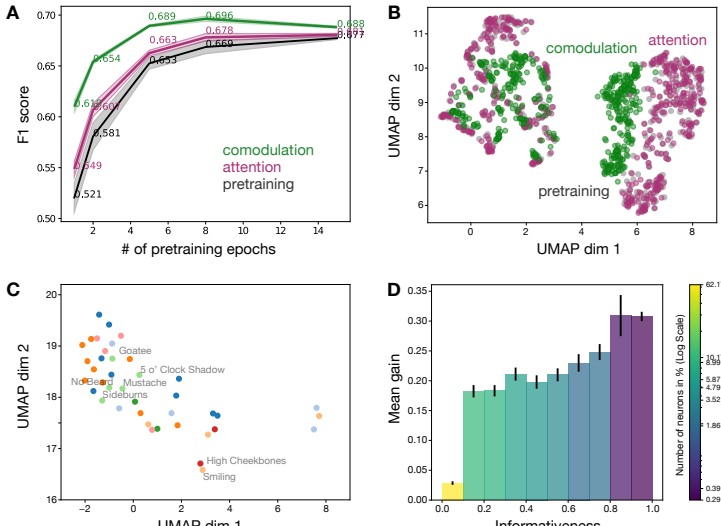

Figure 2: Task fine-tuning in CelebA. **A)** Performance of the 3 models when varying the number of pretraining epochs, the gaps between the models decrease when increasing the pretraining epochs. **B)** UMAP (McInnes et al., 2018) projection of the decoder representation for the 3 models. The embeddings of the attention model are similar to those from the pretrained model, whereas the co-modulation embeddings demonstrate a significant shift. **C)** UMAP projection of the context weights, the controller keeps similar tasks close in the embedding space. **D)** Histogram of the mean gain of neurons grouped by informativeness for one task. The mean gain increases with the informativeness

baseline, with adjustment $p_n = 1$ if a better performance means a lower score and $p_n = 1$ otherwise. We quantify $\Delta_p$ on the validation dataset, as previously done by (Pascal et al., 2021; Ding et al., 2023). Additionally, we report the results on the test data from (Liu et al., 2015). Comodulation performs better than the previous SOTA, ETR-NLP (Ding et al., 2023), by $\Delta_p = 2.8\%$; this is a significantly larger improvement compared to the previous SOTA performance increase (1.2% improvement ETR-NLP vs. max roaming). [1]

**Comodulation learns quickly.** Multi-task learning methods often train the model from scratch and this requires a lot of training epochs to perform well, e.g. Pascal et al. (2021); Ding et al. (2023) both use 40 epochs for CelebA. This is not the case for our stochastic comodulation approach: since the ResNet is already pretrained on Imagenet, we only need to pretrain the full architecture for 8 epochs and then fine-tune for 1. This is consistent with previous observations that pretrained networks can transfer well to other tasks (Yosinski et al., 2014; Donahue et al., 2014; Li et al., 2019; Mathis et al., 2021) and that a good embedding model is helpful to achieve high performance in general. The same has been documented for meta-learning (Tian et al., 2020) and perhaps it also applies to attribute classification. Increasing the number of training epochs reduces the performance difference relative to deterministic gain modulation, but comodulation outperforms alternatives by a large margin in the little training regime (Fig. 2A). We only need 5 epochs of pretraining to fine-tune with comodulation and beat state of the art. The decreasing gap between the models could be due to the network weights overfitting to the training set, and the representations of the different tasks becoming increasingly entangled and harder to fine-tune by gain modulation.

**Mechanism.** In Fig. 2 B, we show UMAP (McInnes et al., 2018) embeddings of the decoder's activity for one task. Somewhat unexpectedly, the embeddings due to deterministic gain modulation do not differ substantially from those of the pretrained model. On the other hand, comodulation shifts representations by a larger margin. The embeddings tend to aggregate more, as the random noise within class is suppressed. The controller's representation of tasks is also meaningful (Fig. 2C): the

---

[1]Unlike other methods, precision is lower than recall. This means that the comodulation tends to have a positive bias, perhaps due to the positive increase in gain.

UMAP projection of the controller outputs for the 40 tasks, grouped into 8 distinct superclasses, as done in (Pascal et al., 2021), by color seems intuitively sensible, e.g. tasks concerning facial hair aggregate together.

The original theory of comodulation predicts that the stochasticity in the modulator should target preferentially task-relevant neurons in the encoding layer and then propagate downstream. This is indeed the case in our network, as the gains of the decoder layer are larger for highly informative neurons (Fig. 2D), where informativeness is measured by a standard $d'$. This suggests that when given the opportunity to use stochasticity for fine-tuning, the network will find the same kind of solution as that seen in the brain.

Table 1: Comparisons of state-of-the-art methods on the CelebA dataset, adapted from (Ding et al., 2023). The best result for each metric is shown in bold. Bottom 3 lines are the results on the test set.

| Method (ResNet18) | #P (M) | 40 facial attributes (tasks) | | | |
|---|---|---|---|---|---|
| | | Precision (↑) | Recall (↑) | F-score (↑) | $\Delta_p$ (↑) |
| Hard sharing | 11.2 | $70.8_{\pm0.9}$ | $60.0_{\pm0.3}$ | $64.2_{\pm0.1}$ | 0.0% |
| GradNorm ($\alpha = 0.5$) (Chen et al., 2018) | 11.2 | $70.7_{\pm0.8}$ | $60.0_{\pm0.3}$ | $64.1_{\pm0.3}$ | -0.1% |
| MGDA-UB (Sener & Koltun, 2018) | 11.2 | $71.8_{\pm0.9}$ | $57.4_{\pm0.3}$ | $62.3_{\pm0.2}$ | -2.0% |
| Atten. hard sharing (Maninis et al., 2019) | 12.9 | $73.2_{\pm0.1}$ | $63.6_{\pm0.2}$ | $67.5_{\pm0.1}$ | +4.8% |
| Task routing (Strezoski et al., 2019) | 11.2 | $72.1_{\pm0.8}$ | $63.4_{\pm0.3}$ | $66.8_{\pm0.2}$ | +3.9% |
| Max roaming (Pascal et al., 2021) | 11.2 | $73.0_{\pm0.4}$ | $63.6_{\pm0.1}$ | $67.3_{\pm0.1}$ | +4.6% |
| ETR-NLP (Ding et al., 2023) | 8.0 | $73.2_{\pm0.2}$ | $64.8_{\pm0.3}$ | $68.1_{\pm0.1}$ | +5.8% |
| Hard Sharing (ImageNet Pretrained) | 11.3 | $74_{\pm0.4}$ | $63.5_{\pm0.5}$ | $66.9_{\pm0.3}$ | +4.8% |
| Attention | 11.3 | $\mathbf{74.9_{\pm0.1}}$ | $63.7_{\pm0.3}$ | $67.8_{\pm0.2}$ | +5.7% |
| Comodulation | 11.3 | $66.3_{\pm0.2}$ | $\mathbf{74.3_{\pm0.2}}$ | $\mathbf{69.6_{\pm0.1}}$ | +8.6% |
| Hard Sharing (ImageNet Pretrained, Test set) | 11.3 | $74.3_{\pm0.5}$ | $62.2_{\pm0.6}$ | $65.7_{\pm0.4}$ | 0.0% |
| Attention | 11.3 | $\mathbf{74.7_{\pm0.2}}$ | $62.8_{\pm0.2}$ | $66.9_{\pm0.1}$ | 1.1% |
| Comodulation | 11.3 | $66.4_{\pm0.4}$ | $\mathbf{73.4_{\pm0.7}}$ | $\mathbf{68.9_{\pm0.11}}$ | +4% |

## 4.2 IMAGE CLASSIFICATION

**Setup.** To gain insights into our model's functionality, we turn to a slightly simpler dataset, CIFAR-100 (Krizhevsky et al., 2009), as an experimental sandbox. This includes 60,000 $32 \times 32$ color images from 100 different classes. These classes can be grouped into 20 superclasses, making it a suitable data set for both fine- and coarse-grained classification. Since these images are smaller than the ImageNet images that the features extractor was pretrained on, we cut the last ResNet block to remove one step of spatial downsampling, while keeping the rest of the architecture as described above. First, we train this network to classify the 20 superclasses until convergence ("pretraining"). Second, we fine-tune the controller and a new output to handle the fine-grained 100 classes. It is important to note that this multi-task setup is qualitatively different from the CelebA task. If in CelebA each image had the potential of being used across 40 tasks, here one image belongs to a single class output. The goal is to take the pretrained representations, where images from different fine-grain classes may have been lumped together and morph them to encourage better class level separability. This is a sufficiently large departure from the original setup that it may lead to different mechanistic solutions. It does seem in some sense more in the spirit of what context dependence and perceptual learning are thought to achieve biologically.

We consider two architectural variations of the top three modulation-relevant layers. The 'base' version includes two convolutional layers and one MLP layer as in the previous section. The 'residual' version includes additional residual connections between each layer starting from the Resnet features until the decoder (see Fig. S1), where the convolutional layers learn residuals instead of unreferenced functions (He et al., 2016). This also changes the effects of the controller, as not all information reaching the decoder layer is comodulated. In out experiments, the learning rate was independently optimized for pretraining and fine-tuning in each architecture, although these optimal learning rates were relatively consistent across all variants. Since using co-modulation during training achieved better accuracy, see table S1, we used this approach for all subsequent experiments.

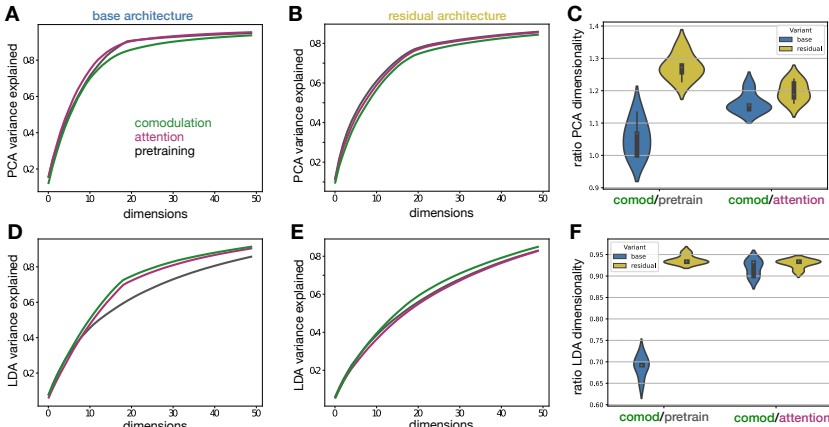

Figure 3: PCA and LDA explained variance. **A)** Variance explained as a function of the number of PC dimensions for base architecture. **B)** Same as A) for the residual architecture. **C)** Violin plot of the ratio of the number of dimensions needed to reach an 80% variance explained threshold by comodulation versus the pretrained model or attention. **D-F)** Same metrics as in A-C) but specifically in the task-relevant axes, as measured with LDA.

**Residual connections aid comodulation.** We first look at the test classification accuracy of the two versions. The residual model fine-tuned with comodulation achieved the highest accuracy, outperforming the best attention-based model by 1.8%. Interestingly, the best-performing model for attention is the base model (68.9% vs. 67.4%), with the base model with comodulation only improving by 0.7% over the attention (69.6% accuracy). The reason for these results are not completely clear. The poor fine-tuning performance of attention in the residual architecture may be due to the fact that skip connections diminish the controller's influence on decoder activity. The skip connections might also change the informativeness statistics of responses in a way that aids comodulation.

**Comodulation shrinks decoder noise specifically in the discriminative manifold.** We measured the structure of the noise in the decoder manifold in two different ways. First, to capture the overall spread of the representation we used the PCA spectrum of the gain-modulated decoder activity in response to all images in the test set (Fig. 3A,B). Second, as a measure of variability specifically affecting class discrimination, we used linear discriminant analysis (LDA) on the same vectors given the corresponding fine class labels (Fig. 3D,E). While the differences are subtle, we found that attention shrank the overall variability more[2], but comodulation leads to lower dimensional representations in the task-relevant submanifold, leading to better class separability overall. This effect was robust across networks (5 seeds). The ratio of the numbers of dimensions needed to explain 80% of the variance is larger than 1 for PCA (Fig. 3C), and systematically lower than 1 for LDA (Fig. 3F). The only architectural difference in these results was in terms of the effects of fine-tuning relative to pretraining, while the relative comparison between stochastic and deterministic gain modulation was consistent for both base and residual models.

**Comodulation improves confidence calibration.** Confidence calibration is an important aspect of classification (Guo et al., 2017). A well-calibrated model has confidence levels that accurately reflect the actual errors that it makes, ultimately making the model more reliable . In other words, when a network predicts class C with a probability of 0.4, the probability that the network is correct should be 0.4. We used reliability diagrams to quantify the confidence calibration (shown in ref-fig:calibrationA for the base model), where any deviation from the diagonal marks a miscalibration (gaps in red). The Expected Calibration Error (ECE) (Naeini et al., 2015) is a metric that quantifies the net degree of model miscalibration, with 0 corresponding to perfect calibration and high values signaling poor calibration. Fig. 4B compares the ECE values for attention and comodulation across 5 networks. Across all instances, calibration was significantly better for comodulation, suggesting that the injected noise also helps assess the relative reliability of different decoder features. This is

---

[2]Overall variability was measured as lower number of PCs needed to explain a given amount of variance.

true even in scenarios where the performance of deterministic and stochastic modulation is comparable. Moreover it is known that better accuracy does not necessarily imply better ECE (Guo et al., 2017); e.g. the older CNN LeNet (LeCun et al., 1998) has better ECE despite lower accuracy on the CIFAR-100 dataset compared to ResNet. Overall, these results that the use of stochastic modulation endows the system with better confidence calibration, separately from its benefits on classification performance.

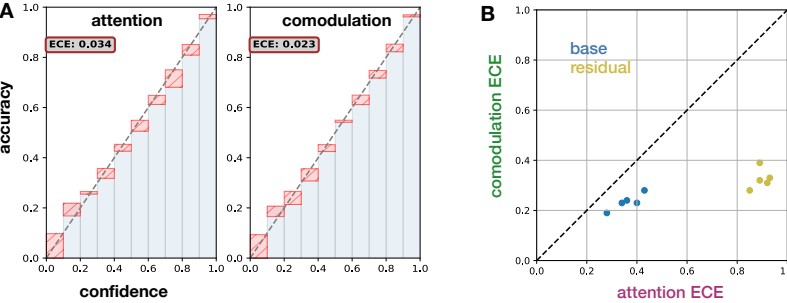

Figure 4: **A)** Example model calibration for the base architecture using either stochastic or deterministic gains. Reliability diagram measures deviations from the perfect confidence calibration which matches true model accuracy (dashed line); red shaded areas mark gaps between the two, which are summarized in the ECE statistic, the smaller the better. **B)** Scatter plot of ECE values of the two models for the 5 seeds.

**CIFAR-100 embeddings.** At the mechanistic level, the difference between comodulation and attention lies in the decoder gain. We thus wondered how the decoder activity differed in the two models. We found that decoder embeddings of the "Vehicle 2" superclass, projected with UMAP, were more tightly grouped for comodulation while preserving separable structure across classes (Fig. 5). While less clear than in the CelebA example, comodulations leads to more substantial shifts of the embeddings compared to deterministic gains, which also explains why comodulation has higher accuracy; for example in the residual models, samples of the 'tank' label are near the cluster of the 'lawn-mower' in the attention embeddings, but in the comodulation there is only one, and is further away from the other class's center.

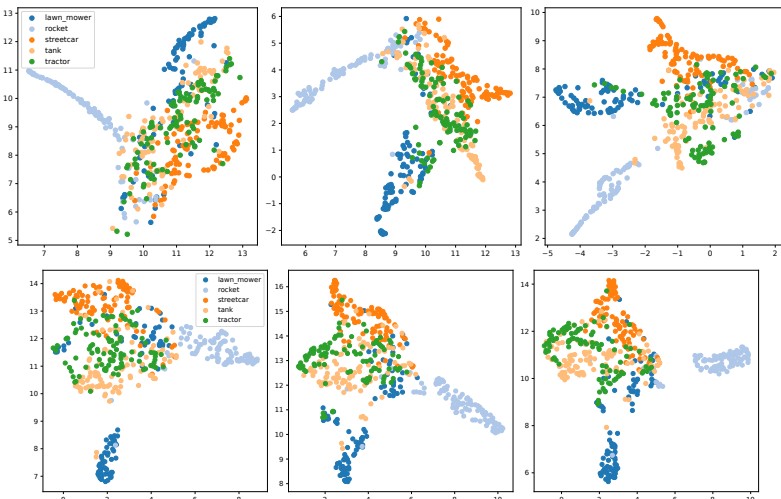

Figure 5: UMAP projection of the different models. Top: base; Bottom: residual. Columns from left to right: Pretrained, Attention, Comodulation.

**Comodulation targets informative neurons.** As in the CelebA case, we wondered to which extent the trained networks converged to the same kind of solution predicted by the original theory. We again assessed how gain change as a function of informativeness (Fig. 6), where informativeness is defined as

$$\text{info}_n = \frac{\partial o}{\partial a_n} \times a_n,$$

where $o$ denotes the activity of the ground truth label output neuron, and $a_n$ is the activity of the output neuron (Baehrens et al., 2010; Simonyan et al., 2013). Indeed, higher informativeness leads to higher gains, reinforcing the previous results on CelebA. Interestingly, the fraction of informative neurons differs across architectures as can be seen in Fig. 6, which may explain why comodulation does better in the residual version.

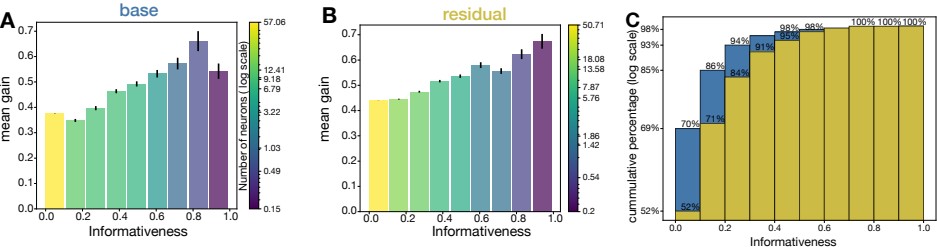

Figure 6: **A)** Gain modulation as a function of informativeness for base architecture. **B)** Same for residual. **C)** The cumulative distribution of informativeness values for the two architectures.

## 5  DISCUSSION

Neuronal representations of the sensory world need to stably reflect natural statistics, yet be flexible given contextual demands. Recent experimental and theoretical work has argued that stochastic comodulation could be an important part of the mechanistic implementation of this adaptability, quickly guiding readouts towards task-relevant features (Haimerl et al., 2021). Nonetheless, the computational power of stochastic-based gained control mechanisms remained unclear. Here, we tested the ability of comodulation to fine-tune large image models in a multi-task learning setup. We found that co-modulation achieves state-of-the-art results on the CelebA dataset. It also provides systematically better model calibration relative to deterministic attention on CIFAR-100, with comparable or better classification performance. Finally, at the level of the resulting neural embeddings, we found that comodulation reshapes primarily the representation in the task-relevant subspace. This suggests that stochastic modulation might be a more effective mechanism for task-specific modulation than deterministic gain changes and a computationally viable candidate for contextual fine-tuning in the brain.

In terms of computational effort, we found that our approach requires 5 times fewer epochs of learning compared to alternatives and often can use off-the-shelf pretrained architectures as building blocks, such as a ResNet as backbone, and a controller trained for attention. It remains unclear under which conditions adding modulation when training the controller is beneficial, but the outcome likely reflects a trade-off between bias (due to using a mismatched gain modulator) and variance (due to the additional stochasticity in the gradients).

The location of the encoding layer is expected to play a critical role in the quality of task labeling. In the case of abstract category labels task relevant features may segregate relatively late in the representation which explains why comodulation worked best at the top of the visual processing architecture. Biologically, the task-relevant features may be distributed more broadly across architecture, requiring the controller circuitry to target different representational layers as a function of context (implementable with some form of group sparsity on the controller outputs). Future work will need to explore more broadly the effects of architecture and learning across a wider set of tasks.

## REPRODUCIBILITY STATEMENT

All experiments were implemented with Python 3 and Pytorch (Paszke et al., 2019). Every experiment was repeated with 5 seeds, which changed the initialization of the network, and details of the training process. Given these seeds, running an experiment always produces the same output. We will submit the code as a link to an anonymous repository at discussion time. The CIFAR-100 and CelebA data are easily accessible online.

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

## A    EFFECT OF COMODULATION

We write the effect of comodulation on an MLP of three layers with ReLU activations, these can be mapped easily to convolutional layers. The first layer is the encoder, the second a processing layer, and the last the decoder. We focus on two cases, we show that comodulation without using bias does trivial computations where the decoder gain is equal to the neural activity of the decoder. We then show the effect of comodulation when the layers have biases. We denote with $z_{lk}$ layer l's activity for a modulation noise $m_k$. Let $z_e$ be an arbitrary neural activity for the encoder, propagating it with a random positive modulation noise until the decoder gives:

$$\hat{\boldsymbol{z}}_{ek} = \boldsymbol{z}_e m_k \tag{5}$$

$$\boldsymbol{z}_{pk} = \text{rectifier}(\boldsymbol{W}_p \hat{z}_{ek} + \boldsymbol{b}_p) \tag{6}$$

$$\boldsymbol{z}_{dk} = \text{rectifier}(\boldsymbol{W}_d \boldsymbol{z}_{pk} + \boldsymbol{b}_d) \tag{7}$$

$$\boldsymbol{z}_{dk} = \text{rectifier}(\boldsymbol{W}_d \text{rectifier}(\boldsymbol{W}_p \hat{z}_{ek} + \boldsymbol{b}_p) + \boldsymbol{b}_d) \tag{8}$$

$$\boldsymbol{z}_{dk} = \text{rectifier}(\boldsymbol{W}_d \text{rectifier}(\boldsymbol{W}_p (\boldsymbol{z}_e m_k) + \boldsymbol{b}_p) + \boldsymbol{b}_d) \tag{9}$$

$$\boldsymbol{z}_{dk} = m_k \text{rectifier}(\boldsymbol{W}_d \text{rectifier}(\boldsymbol{W}_p \boldsymbol{z}_e + \frac{\boldsymbol{b}_p}{m_k}) + \frac{\boldsymbol{b}_d}{m_k}) \tag{10}$$

Using stochastic modulation along with biases brings variability in the decoder activity by changing the value of the biases.

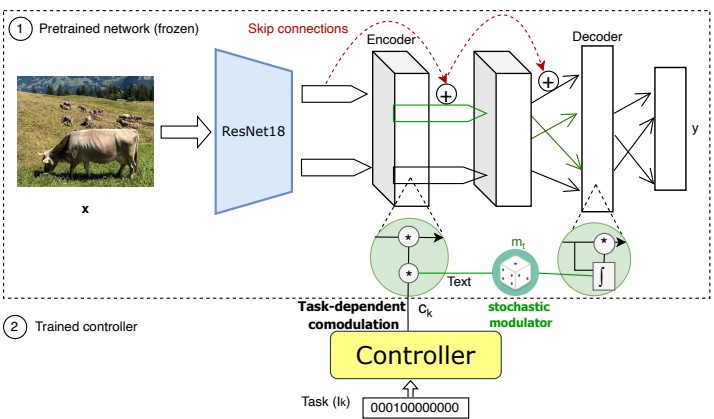

Figure S1: Variation of the architecture with additional residual connections.

Table S1: Accuracy of the different models on CIFAR-100.

|  | Base Model | Residual |
| --- | --- | --- |
| Output weights only | 54.5 | 62.8 |
| Attention | 68.9 | 67.4 |
| Comod test time, one gain per task | 66.4 | 69.7 |
| Comod test time | 68.6 | 69.4 |
| Comodulation, one gain per task | 67.2 | 69.9 |
| Comodulation | 69.6 | 70.7 |

## B  Network with residual connections

## C  Classification accuracy on CIFAR-100

We show in table S1 with the accuracies of the different models on the CIFAR-100 dataset. We also plot in Fig. S2 the difference in accuracy between comodulation and attention of the two models while evaluating during the first 50 batches of finetuning. The dynamics of the two models are opposed, at first the residual comdulations is worse than the attention, but it then flips around, whereas for the base model, the comodulation is better at first.

## D  Residual connections make the gain sparser

We show in Fig. S3 the number of gains computed on the whole test set that are close to zero, with different thresholds defining the closeness, on one seed. As we can see using residual connections increases the number of gains that are close to 0, this means that with residual connections there are more decoder neurons that stay unchanged when the encoder is exposed to different modulation noise. This means that the gain labels fewer neurons as task-informative. Having less informative neurons is the regime in which comodulation was theoretically shown to work better, which gives a hint as to why the residual models work better with comodulation than the base model.

We show here the effect the residual connections have on the sparsity of the gain.

## E  Using a sigmoid on the controller

We also tried using a sigmoid on the controller's output, such that the controller would behave as attention, which can just choose to attend to or not to certain features. The range of the sigmoid is

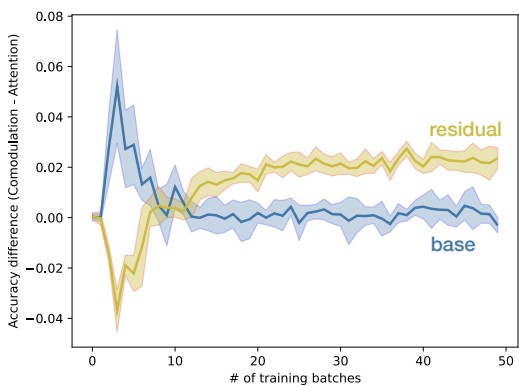

Figure S2: Difference in evaluation accuracy between comodulation and attention of the two models.

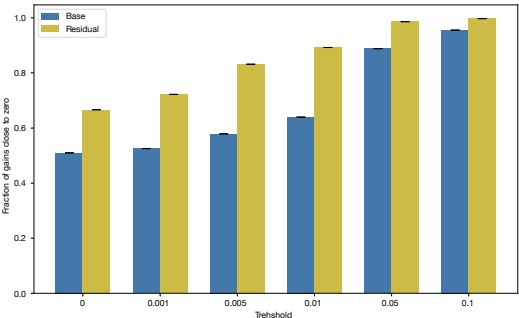

Figure S3: Number of gains before normalization close to 0 with increasing threshold for closeness to 0.

Table S2: Accuracy of the different models on CIFAR-100 when using a sigmoid on the controller.

|  | Base Model | Residual |
|---|---|---|
| Output weights only | 54.5 | 62.8 |
| Attention | 67.1 | 66.3 |
| Comod test time, one gain per task | 66.4 | 69.3 |
| Comod test time | 67 | 69 |
| Comodulation, one gain per task | 67.1 | 69.3 |
| Comodulation | 67.8 | 70.6 |

[0,1] which is a good way of implementing attention in artificial neural networks. As we can see in table S2, there is a more significant gap in the difference between attention and comodulation, it is 3.5% as opposed to only 1.4% in the unbounded controller. We did not use a sigmoid in the main text because it did not perform as well in the CelebA task, to have a more uniform method, we chose not to conduct experiments with it.

## F  A FIXED GAIN IMPROVES ROBUSTNESS AGAINST CORRUPTIONS

We also tested the robustness of our method against common corruptions as defined in (Hendrycks & Dietterich, 2019). Specifically, we use 6 corruptions and vary their degree to go from uncorrupted to high-intensity corruptions. We found that when computing the bias in an online fashion the comodulation exhibits the same level of robustness as attention. However, when computing the gain for each image in the training set, then averaging them per task, and using those gains when testing, comodulation achieves a higher degree of robustness. We plot this in Figure S4. As we can see for every model, fixing the bias helps in having better robustness to perturbations, even though the difference in accuracy is lower when uncorrupted. We did not use this model for every experiment as the accuracies with fixed gains are lower as can be seen in S1.

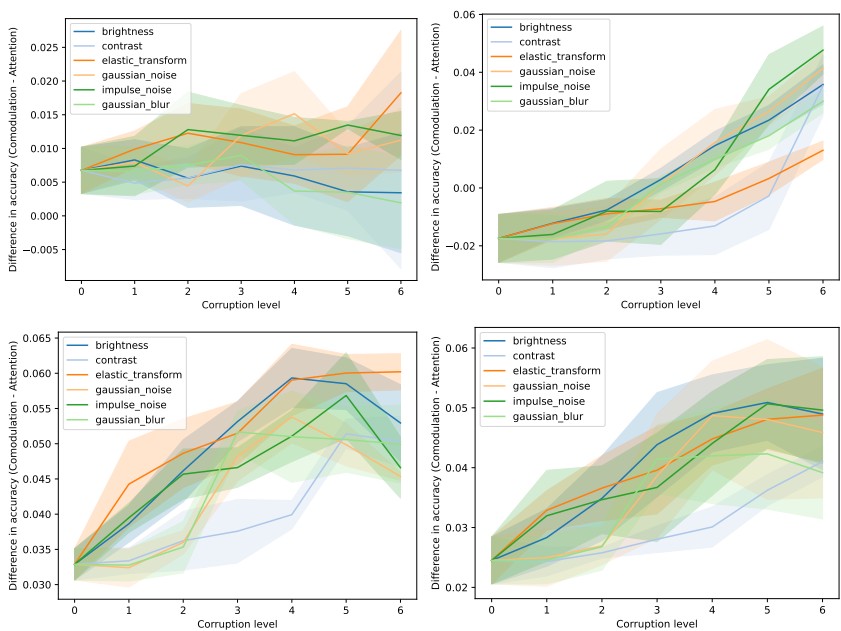

Figure S4: Difference in accuracy between comodulation and attention for the three models. Left is without fixing the gains, right computes the gains on the training set. Top is base model, bottom is residual.

