# OpenReview forum: "Task adaptation by biologically inspired stochastic comodulation"
_ICLR.cc/2024/Conference — Submitted to ICLR 2024_

### Official Review · Reviewer_d7DF · 2023-10-30

**Soundness:** 2 fair
**Presentation:** 2 fair
**Contribution:** 3 good
**Rating:** 5
**Confidence:** 3

**Summary:**

This paper presents a neuroscience-inspired method to enhance multi-task adapation in deep learning. In particular, it proposes stochastically co-modulating neurons' gains based on their task relevance in multi-task adaptation problem. The methodology involves mechanisms of stochastic comodulation and neural architectures design. Numerical experiments have been conducted in CelebA and CIFAR-100 datasets, where the experimental settings were slightly different. The experimental results demonstrate the effectiveness of the proposed method. The paper also investigates the mechanisms supporting this improvement by a range of visualizable analysis. Overall, the paper is an interesting attemptation in the adaptation in multi-task learning, while there remains some space to improve.

# Post-rebuttal
While I appreciate the authors for the responses and clarifications, they do not solve my concerns (e.g., no addtional experiments on ImageNet or CoCo). Therefore, I will keep my original recommendation.

**Strengths:**

1. The proposed stochastic comodulation method is simple and effective, as well as faster in convergence.

2. The paper provides comprehensive and interesting analysis about the learned model properties (Figure 3-6).

3. This work shed light on understanding stochasticity nature of neural representations in the brain.

**Weaknesses:**

1. While it is interesting to see a bio-inspired algorithm, I found the motivation is a bit hard to follow without the background knowledge about "co-modulation" in the brain. There is somehow a gap between the neuroscientific findings and the proposed deep learning methodology in this paper (I have not read the preset papers (Haimerl et al. 2019; 2022)).

2. The novelty of this paper is unclear in presence of (Haimerl et al. 2022). See the questions below.

3. The testbeds could be more extensive. See the questions below.

Minor:
- Colors in Figure 1, 2, 3,5 is not friendly to color-blind readers (especially red-green), consider using different marker/line styles instead.
- Plot labels in Figure 5 are too small and hard to recognize on printed paper.
- REPRODUCIBILITY STATEMENT: Pytorch -> PyTorch
- Is there any reason that Haimerl et al. 2022 used the term "co-modulation" but this paper use "comodulation"?

**Questions:**

1. The paper wrote "We extend the model of stochastic comodulation presented in Haimerl et al. (2019; 2022) ...". Could the authors explain in more details about the relationship between the current paper and (Haimerl et al. 2022), including methodology, task setting etc. ?

2. I also wonder how the presented stochastic comodulation method perform on more challenging dataset such as ImageNet and COCO. Is there any reason that ImageNet was not tested?

3. It might be interesting to investigate whether stochastic comodulation could mitigate the notorious catastropic forgetting problem in continual/lifelong learning, since deep learning AIs are known to be suffering much more from catastropic forgetting than biological agents. Do the authors have any prelimiary results or thoughts about this?

4. What are the relation between the proposed methods and LoRA / control net, as these methods freeze the pretrained-model and conducts adaptation with additional network of fewer parameters?

---

> ### Author Response · Authors · 2023-11-22
> **Reply reviewer d7DF**
>
> Thank you for the feedback.
>
> Key idea of co-modulation: we have revised the methods text to make clear what the starting point of the is from a biological perspective and given the stochastic modulation recent theory, and what are novel elements unique to our solution.  Briefly, the previous work had proposed a mechanism for changing the readout of a sensory representation in a task specific way, based on task-relevance tags provided by targeted stochastic modulation. What is completely new in our paper is 1) the controller that dynamically decides which neurons to stochastically comodulate as a function of the task and 2) the nature of the computation: everything in the work before involved one task at the time whereas here we focus on different versions of multitask learning, in which the same inputs need to be classified along a large set of decision axes (in CelebA) or representations built for coarse categorization needs to be refines for high precision discriminations (in Cifar-100). Finally the computational relevance of comodulation has only ever been demonstrated in very simple toy tasks, so our work provides an opportunity to test the computational limits of the idea in more substantial machine learning problems (see also reply to reviewer 8Tjw).
>
> Typos and co: thanks for catching those. We use ‘comodulation’ as it is shorter, but ‘co-modulation’ would also be fine.
> Color scheme: we have now revised all the figures to use a published color-blind visible palette. Hopefully that helps improve visual clarity.
>
> Itemized answer to questions:
> 1. Similarities: the functional form of the gain modulation in the encoder layer is identical, the decoder gain computation procedure is in the same spirit, although different in the details. Differences: architecture, in particular the presence of the controller, nature of the tasks, the stages of the training procedure (see also general reply above). The insights about the computational roles of comodulation for task flexibility and the mechanistic insights about how it is achieved at the decision stage are also unique to our work.
>
> 2. Using ImageNet and COCO:  The main purpose of comodulation is to tweak representations in a network trained on multiple tasks, so as to make the representations task-dependent. We thus need datasets that have an intrinsic multi-task logic to them. This is not the case for either ImageNet or COCO, at least not in their regular use. ImageNet would be thought of as a scaled up version of Cifar100, but without the coarse to fine labeling. COCO does have multiple tasks, but they are usually not learned together, and when they are, the models used have a large amount of unshared parameters. Our decoder formulation would likely not be useful in that setup.
>
> 3. Ours is not strictly a continuous learning setup, but it does have a potential advantage in that it structurally segregates common useful regularities in the data (in the backbone) from unique task specific adaptations (in the controller). It is not inconceivable that this separation could provide more robustness to across task interference, but we have not explicitly quantified this effects. It is however an idea worth further thinking on.
>
> 4. LoRA is low-rank adaption of large language models (https://arxiv.org/abs/2106.09685) , control net (https://arxiv.org/pdf/2302.05543.pdf) is a method for adapting diffusion models (hopefully we got the references you intended here). They mark bigger departures from the type of task adaptation intended here. Similarities: Both aim to fine-tune an existing representation. Differences: Lora optimizes new sets of weights which linearly modify the frozen weights in the attention heads, while control net learns new convolutional layers which modify the internal representations in some layer of the UNet. Both methods modify the representations at multiple stages of processing, whereas ours only directly modifies one layer. Neither are meant for multi-task learning, which is the main focus in this paper.

---

### Official Review · Reviewer_iLtH · 2023-10-31

**Soundness:** 2 fair
**Presentation:** 2 fair
**Contribution:** 2 fair
**Rating:** 3
**Confidence:** 3

**Summary:**

This paper introduces biologically inspired stochastic co-modulation as a way to modulate the context of a multi-task learning framework and improve performance. Neural networks are fine-tuned with stochastic modulation gains. The authors show an increase in performance compared to non-stochastic gains, and claim that the networks train more efficiently when stochastic co-modulation is present.

**Strengths:**

This paper connects recent work investigating context-dependent stochastic co-modulation in biology to the multi-task learning setting, showing that it can be beneficial as a way to fine-tune the network performance. This is generally of interest to both neuroscience audiences and machine learning practitioners. I also found the experiment on CIFAR-100 (training on the 20 superclasses and fine-tuning on the 100 classes) to be an interesting way to connect perceptual learning experiments in neuroscience. The authors also demonstrated a notable improvement using stochastic co-modulation compared to the other tested models, particularly in terms of the recall score.

**Weaknesses:**

1. **Clarity of writing** Overall, I found the writing of the paper quite hard to follow. A lot of ground is covered by their experiments, and the motivation is often unclear (for instance, explaining the two versions of the experiment described on page (6) for the “Image Classification” experiment). Additionally, in many places of the paper, there are ad-hoc observations that are not fully explained, for instance, in the footnote on page (5) or the explanation about “network weights overfitting too much on the training set” on page (5). There are also terms that are not well defined, for instance when referring to the “reliability diagrams” on page 7, where “reliability” is not explained. These are just a few examples, and I encourage the authors to revise the text to make it more clear for the reader.
2. **Experimental details seem ad-hoc** The training setup seems a little odd. As far as I can tell, for the CelebA experiment, the authors (1) took a pre-trained ImageNet backbone (2) trained the whole network on the full task (3) fine-tuned just the controller parameters. This makes it difficult to compare the “efficiency” of training because there are many stages involved. Can the controller not be trained at the same time as the encoder/decoder (and if so, is this due to training problems or due to some experimental design choices)?
3. **Evaluation Metrics** The sharp drop in Precision for the proposed stochastic comodulation is somewhat troubling, and the resulting explanation seems insufficient. Even if it is unintentional, it makes the metrics used for comparison seem somewhat designed to ensure that the comodulation model is listed as “best.”

**Questions:**

a. Is it typical to first train the entire network on all tasks and then fine-tune it on the same dataset? I am most familiar with work that fine-tunes for new tasks, so just clarifying whether this is a standard choice would be helpful.

b. I’m a bit confused by the sentence “In other words, when a network predicts class C with a probability of 0.4, the probability that the network is correct is 0.4.” in the “Comodulation improves confidence calibration” section on Page 7). Is there a typo? If not, could you explain this in more detail, as I don’t understand how this could be correct as written. For instance, if Class C is only ever encountered <0.01% of the time then the network will be correct with a probability significantly less than 0.4, right?

c. In Table 1, are the comparison “state of the art” methods computed on models that were trained in the same environment that is used in this paper, or are these taken directly from the cited papers? This detail seems important to clarify, as there might be other underlying differences.

d. Could the authors provide further explanation of the deterministic “attention” model? I could not find details of this in the paper. This seems particularly important to explain, given that it is one of the critical comparisons and does best on the “Precision” metric. Is this attention just the controller without a stochastic element, and if not, why is something along those lines not a direct comparison?

e. As a followup to (d), did the tuning of hyperparameters for training etc. for this “attention” model receive as much tweaking as the co-modulation comparison? It is a little puzzling to me that there is very little change in the representations after fine-tuning here, and so I wonder if it is the stochastic co-modulation that is actually helping, or if the “attention” case had hyperparameters that did not behave well with the training setup.

f. On page 4 there is the sentence “...we only use stochasticity to compute the gain, but we use unperturbed representations for decision making.” This doesn’t make sense to me – isn’t the gain perturbing the representations? Or is stochasticity somehow turned off during the testing (and only on during training?)

e. Minor: there should be a comma in “recall, precision and F1-score” on page 5.

---

> ### Author Response · Authors · 2023-11-22
> **Reply reviewer iLtH**
>
> We thank the reviewer for the feedback.
>
> Text clarity: We agree that rushing towards the deadline made the text fall short in terms of clearly explaining some of the details. We have now revised the main text to hopefully address various clarity concerns.
>
> Training procedure: We tested both training scenarios: 1) where the controller is trained jointly with the network, and 2) where the controller is trained separately with the fixed pretrained backbone. We chose to report the second option for consistency with the other experiment (CIFAR-100 setup) and because we found that training the controller separately made the training more stable. Finally, training the controller is fast (you only need to backpropagate gradients up to the encoder) so that overall training time is not substantially increased if done in two stages.
>
> Evaluation metrics: The metrics used were chosen based on the previous literature on multitask learning in CelebA, which are also reported in Table 1. Precision is defined as true positives/(false positives +true positives), thus if the network classifies only one instance as a positive and is correct it will have a precision of 1. Comodulation behaves a little unusual by this metric, for reasons that we don’t completely understand.  Recall measures the amount of true positives/(false negatives + true positives); the two naturally trade-off between each other. The F1 score is the harmonic mean of the two, and quantifies the trade off between recall and precision. F1 is the key metric to look at. We don’t see any reason why the precision properties of comodulation would trivially explain the benefits seen in terms of F1 scores.
>
> Itemized question answers:
> a) Separate phases of training are not typical, it is a design choice that we came to via numerical experimentation (see also training explanation above).
>
> b) Confidence calibration is defined as model predicted confidence matching actual errors made for that with that confidence level. So for the set of images that the model predicts that it should be 0.4 sure about the class label, 40% of those should be mislabeled. We tried to rephrase the text to better convey the idea.
>
> c) Code is only  available for some options (Max roaming, Pascal et al. (2021)) so we used the evaluations previously reported in (ETR-NLP Ding et al. (2023)) Table 4, while matching our evaluation procedure to theirs as much as possible. We use the same batch size, augmentations and optimizer. Worth noting that the code is not available for ETR-NLP Ding et al. (2023), so one cannot be sure that the comparison is completely one-to-one. Nonetheless, we did our best to ensure it would be the case based on the methods described in the paper.
>
> d) Thanks for pointing this out. The attention baseline is indeed the model that you describe, now explained in the text.
> The procedure for hyperparameter tuning was identical for both stochastic and deterministic attention.  So, yes it did receive the same degree of attention (pun intended). The hyperparameter set included  the size of the hidden layer of the controller in the range [32,64,128,256] and the learning rate of the controller, on a log scale from [0.0001 to 0.1]. The pretraining hyperparameters optimization aimed to achieve the best results on the pretraining task in the absence of the controller, as would be the case in practical ML applications.
>
> e) Apologies,  we referenced the wrong equation. What we meant is that the encoder is not stochastically modulated at test time, i.e we only use the stochastic modulator to estimate the decoding gain. We have changed the text to make this clearer.
>
> Typos fixed, thank you for catching those.

---

### Official Review · Reviewer_cKzA · 2023-11-02

**Soundness:** 3 good
**Presentation:** 2 fair
**Contribution:** 3 good
**Rating:** 5
**Confidence:** 4

**Summary:**

The authors propose a novel task adaptation technique for multi-task learning based on task-specific stochastic comodulation of neurons. The proposed approach modulates a pretrained backbone's (ResNet-18) response using iid gaussian noise ($m_t$) for stochasticity, combined with a learned representation of context information ($c_k$) and forwarded to an MLP decoder. For each downstream context $k$, the readout is obtained by scaling the decoder's activations using a context-dependent gain ($g_k$). The above described comodulation process repurposes the outputs from a strong pretrained backbone to be used for a variety of downstream tasks by identifying (and scaling) task-relevant neurons from the backbone. Results on Celeb-A and CIFAR-100 show the merits of comodulation in producing improved performance (as measured by F-score) and prediction confidence calibration. The authors also include analysis of comodulated network's internal representations to show that they learn semantically rich decoder and context representations.

**Strengths:**

+ This submission explores repurposing a pretrained backbone's representation with minimal finetuning and extra parameters to improve performance in diverse downstream contexts. This work is highly relevant in this era of strong pretrained visual backbones, and using their representations to perform multiple downstream tasks.
+ The evaluations are quite rigorous with multiple random intializations of all models used to report variance in performance.
+ Analysis of networks trained with comodulation using CIFAR-100 was really interesting, and it was impressive (although not clear why) that comodulation improves calibration of prediction confidence.

**Weaknesses:**

- The improvements produced by comodulation over attention seem quite marginal and tend to diminish as the number of pretraining epochs increases. It is true that as backbone size increases, pretraining becomes less of a viable option, but the small gains over attention regardless (with few pretraining epochs) makes the work less exciting.
- I found the writing to be clear overall, but felt that the Methods section could be further revised for improving readability. E.g. (1) the reader isn't aware of what $h^{J}_t$ is the first time it appears at the bottom of page 3; (2) is $h^{J-1}_k$ in Eqn. 3 the output of the second encoder layer (i.e. $h^{l+1}_k$? (3) citations appear to be in the wrong format in a few places and needs to be corrected
- Although the attention baseline is repeatedly evaluated in many parts of the paper, I couldn't find a mathematical description of this approach, adding which would improve the clarity of this work.

**Questions:**

Please refer to my weaknesses section above. Clarifying my questions and improving the readability of this work will help improve my rating of this submission.

---

> ### Author Response · Authors · 2023-11-22
> **Reply Reviewer cKzA**
>
> We thank the reviewer for the feedback.
>
> Regarding the magnitude of performance improvements due to comodulation: While it is true that the improvements diminish with increasing training epochs, the results from comodulation and attention are quite different when placed in the context of the previous literature. The attention model F1 sits at 67.8 which is below the previous SOTA of Ding et Al 2023. On the other hand, the comodulation has a F1 of 69.6, which in terms of increases relative to the previous SOTA is a significant jump. To put things in perspective, the two most recent  increases in SOTA on CelebA were 0.5 and 0.8, which compared to our 1.8 are much smaller.
>
> We apologize for the lack of clarity of the text, we have revised it to better explain the methodology and the significance of the results. We have also corrected the typos and fixed the issues with improper formatting of references.
>
> Attention/deterministic gain modulation uses the exact same architecture with a multiplicative gain effect on the encoder and no gain effect on the decoder. The procedure for training and hyperparameter optimizations are shared with stochastic modulation, to keep the comparison as fair as possible. We have clarified these points in the methods.

---

> > ### Comment · Reviewer_cKzA · 2023-11-30
> > **Thank you for your response to our reviews**
> >
> > Dear authors,
> >
> > Thanks for responding to our reviews. My main concerns of the paper still remain after the author response; thank you for enhancing the clarity of the text in the paper. I will retain my score and believe that the paper could be improved further before being accepted at ICLR.
> >
> > Thank you.

---

### Official Review · Reviewer_8Tjw · 2023-11-07

**Soundness:** 4 excellent
**Presentation:** 3 good
**Contribution:** 3 good
**Rating:** 8
**Confidence:** 2

**Summary:**

This paper investigates stochastic modulation as a mechanism for adapting neural circuitry to task in large vision models. Motivation comes from biological neural networks' ability to modulate neural responses in a task-dependent manner, suggesting that this may be effective in artificial NNs and that trying it may shed light on biological NNs.

The method is centered on fine-tuning by comodulation. Extending on prior work, the method begins with a pretrained model (here, image models) and a pretrained controller that maps from task indication to task-dependent comodulation as context weights, which ideally magnify task-informative neurons while inhibiting task-irrelevant neurons. Furthermore, a stochastic modulator generates noise to apply to decoder activations. Models are trained on a primary task then tuned on a different but related task; at this point, only coupling weights change.

Experiments are performed on CelebA (40 binary tasks, some as primary and some as secondary, each with the same input image distribution) and CIFAR-100 (one classification task into 20 superclasses, then a fine-tuning classification task into 100 classes).

The paper finds that comodulation enables convergence in far less training than in other multi-task learning-related fine-tuning methods, though with more tuning, deterministic attention (another mechanism inspired by enhancing certain neurons and suppressing others) performs similarly. Furthermore, the paper discusses that CIFAR-100, residual connections improve comodulation, comodulation shrinks decoder noise, and comodulation improves the model's confidence calibration.

The paper concludes that comodulation can help achieve SOTA results, faster training, and better calibration than deterministic gain modulation in some conditions. It is an open question which specific conditions this applies to. The paper argues that the results also demonstrate stochastic modulation to be a computationally viable candidate for contextual fine-tuning in an animal brain.

**Strengths:**

#### Quality
- Paper is well-motivated by biological systems and a natural problem for ML
- Experimental setup is thorough.
  - Particularly clever: training/fine-tuning division on CelebA vs. CIFAR-100 requiring two different types of shifts of distribution
- Results are laid out and analyzed well. While the main claims are a bit disorganized, they are all backe dup and convincing, even if at a small scale.

#### Clarity
Paper is very well written.


#### Originality
No originality concerns - paper is well-grounded within prior literature.


#### Significance
The paper is contextualized well in both biology and ML literature, which can be difficult. The improvements on various datasets are nontrivial, and compete with similar types of work in ML. Experiments with the fine-tuning done on larger and more complex datasets would be good for sending a solid ML message, but this is already an interesting contribution.

**Weaknesses:**

#### Quality
- Final results might need more grounding to actual neural data to claim that this paper has made stochastic modulation seem more like it might be used in the brain. "Might be a computationally viable candidate" is technically accurate, but the language still feels overall overstated with respect to how much these results mean for the actual brain.
- While this may be a matter of my not knowing prior work, the mechanism isn't clearly anchored in prior work - even after the experiments section, there's only a vague sense of it. There is one exception, where in the methods section it's made clear what the methods are built off of - more of that might be helpful.

#### Clarity
- It really is hard to see in many of the figures why we should be excited about the differences in the curves. Annotating or zooming in to ground them quantitatively would help.
- "Deterministic gain increases" and "calibration of confidence" are brought up in the intro but not defined until the methods/results, which is confusing.
- Figure 1 should be connected to text earlier than Equation 1 - it's hard to understand what on the figure is referring to what in the paper.
- Figure 1: unclear how the stochastic modulator originates and interacts with the decoder. The controller has task info embedded, but Fig 1 + text makes the stochastic modulator look both like it comes from the controller and like it's purely randomly sampled. Which is it?
- Table 1 would benefit from far more annotation - it's a lot to take in right now, and the text only refers to it as a whole (or it refers to specifics that are hard to find and hold in memory).
- Minor: the paper says the task is given as a one-hot, but the figure makes it seem like a binary representation of some kind.

#### Significance
To some degree it's unclear which audience this paper is for - for neuroscientists, there's almost no quantitative grounding in neural data. That said, the paper seems primarily geared toward the ML community - here, more/larger-scale experimentation would help but the results do show clear and consistent improvement. I think this is a smaller but well-scoped paper that points to interesting further research along with making a contribution.


#### Minor
In the "network embeddings" paragraph in page 5, the paper refers to fig 2D; I think it should refer to fig 2C.

**Questions:**

- How exactly is the stochastic modulator signal integrated into the decoder, and how is it generated? Does it come from the controller or a totally random sample? How does that reflect on Fig 1?
- "Maps individual tasks into the context weights" - did you mean "to"?

---

> ### Author Response · Authors · 2023-11-22
> **Reply reviewer 8Tjw**
>
> We thank the reviewer for the feedback.
>
> With regards to the biological relevance of this work: While data from several experimental datasets show that task-dependent comodulation is present in brain activity (e.g., Rabinowits for V4, Haimerl for V1 and MT), the computational significance of these empirical observations has only been demonstrated indirectly, and using very simple tasks. Thus, the main neuroscience-relevant question we address in this paper is “What kind of computations can such a mechanism support? Is it only viable in single toy tasks or does it have enough computational power to help perception and decision making in ways that matter outside the confines of an experiment?” By proving that stochastic comodulation does actually provide computational gains in challenging machine learning tasks we are, at the very least, showing that it would be beneficial for the brain to use it. Our grounding in neural data is qualitative rather than quantitative, but arguably that is true for many successful models in computational neuroscience.
> Finally, although abstract the model does speak to some qualitative features of neural data. Even if the learning part of the process is not biologically plausible in the details, the architectural structure of the model can motivate the experimental investigation of the brain circuitry performing the function of the controller (putatively superior colliculus) and makes predictions about its modulatory effects on visual areas. One should also note that some of the mechanistic details that one would like to know about stochastic modulation are equally mysterious for the deterministic gain modulation supporting what we typically refer to as “attention” in the brain, so these limitations are by no means unique to our model. Hopefully, our modeling results will further spur investigation of the neural mechanisms supporting task flexibility.
>
> In terms of links to the previous literature: we have revised the main text to hopefully spell out more clearly the similarities and differences between the solution presented here and other machine learning approaches (Related work section) and the previous work on biological roles of stochastic modulation (Introduction and Methods).
>
> Visual clarity: we have changed the training range for Fig 1B to make it clearer that comodulation surpasses the pretraining baseline and converges faster than gain modulation. We have also changed the color scheme to make it more discriminable for the spectrum of the readers. Even when the effects look small on the scale of the range of metric values seen across learning, hopefully the summary statistics make the significance of those distinctions statistically clear.
>
> With regards to unclarity of explanations and various typos: we have revised the text to address each of those concerns. We have also corrected the cartoon to accurately reflect the one-hot encoding of the tasks.
>
> The updated methods clarify a little more the role of the modulator at both encoding and decoding time, and the temporal relevance of those processes. Briefly, the modulator injects multiplicative noise in the encoding layer which propagates together with the image information to the decoder layer. The correlations between neural activity and the same modulator change the layer’s gains, which ultimately affect the output. For simplicity, the modulator is iid gaussian although this can definitely be relaxed with no ill effect.
> We have also tweaked the table to hopefully make the results easier to read and interpret, and added more explanation about the metric in the main text.
>
> Overall, even if the immediate biological relevance of the model is limited, we do present a biologically motivated algorithm that improves on state of the art on nontrivial machine learning tasks. Moreover, the idea is counterintuitive enough that one would not have tried something like it were it not for the biological observations.  Such instances of synergistic interaction between neuroscience and machine learning are not that common, which is why we hope that our work is of general interest to the ICLR community.

---

### Meta-Review · Area_Chair_Dnee · 2023-12-06

**Metareview:**

The paper introduces biologically inspired stochastic co-modulation as a way to modulate the context of a multi-task learning framework and improve performance. While the reviewers appreciate the authors for the responses and clarifications, the authors do not solve the reviewers’ concerns on numerical experiments. Most of the reviewers do not support the publication of the paper. Therefore, the paper cannot be accepted at the current stage. Please consider to resubmit the work to other venues.

**Justification For Why Not Higher Score:**

Most of the reviewers do not support the paper.

**Justification For Why Not Lower Score:**

N/A

---

### Decision · Program_Chairs · 2024-01-16

Reject